# Interaction of Lactoferrin with Unsaturated Fatty Acids: In Vitro and In Vivo Study of Human Lactoferrin/Oleic Acid Complex Cytotoxicity

**DOI:** 10.3390/ma14071602

**Published:** 2021-03-25

**Authors:** Anna Elizarova, Alexey Sokolov, Valeria Kostevich, Ekaterina Kisseleva, Evgeny Zelenskiy, Elena Zakharova, Oleg Panasenko, Alexander Budevich, Igor Semak, Vladimir Egorov, Giulia Pontarollo, Vincenzo De Filippis, Vadim Vasilyev

**Affiliations:** 1Institute of Experimental Medicine, 12 Acad. Pavlov Street, 197376 Saint-Petersburg, Russia; anechka_v@list.ru (A.E.); Hfa-2005@yandex.ru (V.K.); ekissele@yandex.ru (E.K.); ezelen@yandex.ru (E.Z.); et_zakharova@mail.ru (E.Z.); sondyn@yandex.ru (V.E.); 2Chair of Fundamental Problems of Medicine, Saint-Petersburg State University, 7/9 University Embankment, 199034 Saint-Petersburg, Russia; 3Research Institute of Physico-Chemical Medicine, 1A Malaya Pirogovskaya, 119992 Moscow, Russia; o-panas@mail.ru; 4Scientific and Practical Centre of the National Academy of Sciences of Belarus for Animal Breeding, 11 Frunze Street, 220072 Zhodino, Belarus; budevich7388100@mail.ru; 5Chair of Biochemistry, Belarusian State University, 4 Nezavisimosti Avenue, 220030 Minsk, Belarus; semak@bsu.by; 6Saint-Petersburg Konstantinov Institute of Nuclear Physics, 1 Orlova Roshcha, Gatchina, 188300 Leningrad Region, Russia; 7Department of Pharmaceutical & Pharmacological Sciences, University of Padua, 5 via Marzolo, 35131 Padua, Italy; giulia.pontarollo@unipd.it (G.P.); vincenzo.defilippis@unipd.it (V.D.F.)

**Keywords:** lactoferrin, oleic acid, tumor growth, anticancer drugs

## Abstract

As shown recently, oleic acid (OA) in complex with lactoferrin (LF) causes the death of cancer cells, but no mechanism(s) of that toxicity have been disclosed. In this study, constitutive parameters of the antitumor effect of LF/OA complex were explored. Complex LF/OA was prepared by titrating recombinant human LF with OA. Spectral analysis was used to assess possible structural changes of LF within its complex with OA. Structural features of apo-LF did not change within the complex LF:OA = 1:8, which was toxic for hepatoma 22a cells. Cytotoxicity of the complex LF:OA = 1:8 was tested in cultured hepatoma 22a cells and in fresh erythrocytes. Its anticancer activity was tested in mice carrying hepatoma 22a. In mice injected daily with LF-8OA, the same tumor grew significantly slower. In 20% of animals, the tumors completely resolved. LF alone was less efficient, i.e., the tumor growth index was 0.14 for LF-8OA and 0.63 for LF as compared with 1.0 in the control animals. The results of testing from 48 days after the tumor inoculation showed that the survival rate among LF-8OA-treated animals was 70%, contrary to 0% rate in the control group and among the LF-treated mice. Our data allow us to regard the complex of LF and OA as a promising tool for cancer treatment.

## 1. Introduction

Despite the apparent progress in the development of medicine, every year more than 14 million people are diagnosed with malignancies. Both the incidence of cancer and oncological mortality keep growing [1]. The International Agency for Research on Cancer reported about 18 million cases of cancer diagnosed in 2018, of which more than half had a lethal outcome [2]. These numbers have a tendency to grow, which is caused by insufficient prophylaxis, early diagnosis and screening, but also by the imperfection of current antitumor therapies.

State-of-the-art oncology largely uses three basic methods, i.e., surgery, radiation treatment and drug therapy, which includes chemotherapy and a number of novel techniques. However, all methods presently used not every time bring forth positive effect which is the ultimate recovery. Moreover, serious side effects are not infrequent, and some approaches to cancer treatment are efficient exclusively at the early stages of the disease.

Perhaps the crucial factor making treatment of malignant tumors particularly difficult is their rapid growth. Moreover, even a relatively small tumor can disseminate with metastases in distant organs/tissues. On account of these factors, a doctor often chooses a drug treatment having toxic effect both on the tumor and upon non-malignant cells afield. Another problem that complicates treatment is the appearance of tumor multidrug resistance. All this brings forth an extremely important task of looking for novel compounds with low toxicity that target and destroy exclusively malignant cells, but have no effect upon normal ones.

One of the promising current approaches is discovering natural molecules which, upon the contact with malignant cells, either cause their death or arrest their propagation. The most efficient seem to be combinations of two and more molecules normally found in the human organism that do not cause such an effect when taken separately. A number of proteins are known to have specific antitumor activity.

It has been shown that lactoferrin (LF), the protein of milk and other secretions, also present in neutrophilic leukocytes, can suppress the growth of tumor cells in cultures and in animal bodies [3]. This effect is explained by the capacity of LF to stimulate the immune system and to increase the number of apoptotic cells, most probably because of the arrest of cytokinesis at G0-G1 induced by LF [4,5,6]. The threshold of tumor cell sensitivity to chemotherapeutic agents can be lowered by LF, as demonstrated in mice bearing either EL-4, Lewis lung carcinoma or B16 melanoma tumors [7,8]. Natural products and bioactive molecules like LF have noticeable advantages as potential pharmacies thanks to their relative availability and normally high content. They also do not cause immune responses, and sometimes their oral intake is possible. Therefore, it makes sense to undertake a profound study of LF as a prospective anticancer drug. It became even more reasonable when a report appeared on the capacity of non-esterified oleic acid (OA) to cause the death of cancer cells after forming a complex with cow-milk LF [9]. OA has somewhat high toxicity among quite a few fatty acids [10]. However, nothing is known about the mechanism(s) of its toxic effect in complex with LF; hence, the investigation of this phenomenon is really topical.

### 1.1. Lipolytic Effect of LF

Treatment of volunteers with LF caused an egress of fatty acids from the depot and a decrease of body mass index and the amount of visceral fat [11]. When cultured adipocytes were treated with LF, the glycerol content in the medium increased, which is likely to result from the disintegration of triglycerides [12].

### 1.2. Anticancer Effect of LF

The first suggestions of LF anticancer activity appeared when it was shown that the LF gene is not expressed or is absent in cancer cells [13,14,15,16,17,18].

In some types of cultured tumor cells, LF stimulated apoptosis and inhibited proliferation by arresting the transition from the G1 to S phase [19,20,21]. Right at that moment, LF inhibits cyclin-dependent kinases and increases the level of p21, the cell cycle inhibitor, but also sustains retinoblastoma protein (pRb) in its hypophosphorylated form, which most likely arrests the cell cycle in breast cancer cells [21]. Recombinant LF efficiently inhibited both metastatic and non-metastatic breast cancer cells by arresting the cell cycle at the S phase [22], and in squamous cell culture it induced a dose-dependent arrest of cell growth at the G0/G1 phases [6]. Neutrophilic LF activates the synthesis of the tumor suppressor p53 using the NF-κB-dependent pathway [23].

LF is believed to increase the synthesis of tumour necrosis factor receptor (Fas) in colon mucosa both at the early and late stages of carcinogenesis [24]. Particularly enhanced expression of Fas increases the capacity of target cells to bind the Fas ligand exposed at the membrane of natural killer (NK) cells, which induces apoptosis. Interestingly, Se- and Fe-saturated LF more efficiently induced apoptosis in cultured colon tumor cells [25]. Using animal models allowed researchers to show that LF inhibits angiogenesis, in particular, that induced by vascular endothelial growth factor [26,27].

A bulk of evidence was collected in favor of the in vivo protective effect of LF against chemical carcinogenesis in a large number of organs [28,29,30,31]. Clinical studies demonstrated significant growth retardation of colorectal adenomatous polyps in patients receiving LF from cow milk [32]. Its antimetastatic effect was shown in mice with colorectal carcinoma Co 26Lu, perhaps due to the enhancement of immunity, as judged by an increased content of CD4+, CD8+ and NK-cells in the animals’ blood [33]. A similar effect was documented earlier for human LF [34].

When LF is considered as a prospective medicinal agent, its very important feature should not be neglected, i.e., this protein is able to cross the blood-brain barrier and deliver an anticancer drug to the cerebral tissue [35,36].

### 1.3. Complexes Formed by LF and Low-Molecular Substances

Treatment of malignant tumors is not infrequently based on interactions of LF with low-molecular compounds [37,38,39,40,41,42,43]. The conjugation of LF with low-molecular antitumor drugs increases their solubility in water [39]. Besides, a rich vascularization of tumors provides accumulation of such conjugates in malignant cells [40], which results in the gradual release of the drug from the complex. Among the low-molecular anticancer compounds studied so far is temozolomide (TMZ), used for the treatment of malignant gliomas [41]. Nanoparticles formed by LF and TMZ more efficiently cross the blood–brain barrier and reach the tumor cells, where they are retained for a longer period [37]. This allows for treatments to reduce the dosage and avoid some of the side effects of TMZ.

Another example is the complex of LF with 5-fluorouracil (5-FU), the fluorinated nucleotide, which undergoes intracellular transformation into 5-fluorodesoxyuridine monophosphate, an inhibitor of thymidine synthase. This effect causes an arrest of DNA replication, after which rapidly dividing tumor cells switch to apoptosis [43]. Nanoparticles of LF conjugated with 5-FU were obtained for treatment of melanoma [38]. Testing on cultured mouse melanoma B16F10 showed that 5-FU conjugated with LF has cytotoxicity 2.7 times higher than 5-FU alone.

### 1.4. Complexes of Milk Proteins with Oleic Acid

The first complex formed by α-lactalbumin and OA was described some 25 years ago [44,45] and its antitumor activity was revealed [46]. Depending on the biological species that served the source of α-lactalbumin the complexes were termed Human/Bovine/Camel Alpha-lactalbumin made lethal to tumor cells (HAMLET/BAMLET/CAMLET). Along with α-lactalbumin, OA was shown to make complexes with ß-lactoglobulin, termed BLAGLET [47], with equine lysozyme, termed ELOA [48,49], and with other proteins, all of which displayed cytotoxicity towards tumors mostly due to their capacity to initiate apoptosis [50,51,52,53,54,55,56,57]. The capacity to form such complexes is likely to be a common feature of all partly unfolded proteins that have sufficient number of hydrophobic sites to bind fatty acids.

#### Complex of LF with OA

So far, the complex formed by LF and OA has not been featured in detail. An in vitro study revealed its cytotoxic activity in tumor cell lines, such as HepG2, HT29 and MCF-7, where the complex showed the dose-dependent antitumor effect [9]. Since LF itself does not cause apoptosis, the notion that a proapoptotic mechanism of the complex is realized via OA seems quite reasonable. Recently we demonstrated that the complex between human LF and OA interacts with the chromatin of the isolated nuclei of HeLa cells, and thus, affects the chromatin structural organization [58]. However, the overall mechanism of cytotoxicity towards tumor cells displayed by the complex between LF and OA remains a moot question.

### 1.5. Cytotoxicity of Oleic Acid

Non-esterified fatty acids are toxic towards a variety of cells [59,60,61]. OA is not an exception; therefore, it can be regarded as a prospective anticancer agent [10]. Indeed, added to the J774 cell line (murine macrophages), S91 (murine melanoma cells) or fresh human lymphocytes, OA (200 µM) caused DNA fragmentation [59,60,61]. Meanwhile, the same OA concentration had no cytotoxic effect on murine melanoma B16-F0 or human melanoma SK-Mel 23 and SK-Mel, which shows that its toxicity depends on cell type [60]. Interventions of OA into the genetic machinery of tumor cells were also reported in a number of studies. In one of those, OA was shown to inhibit 5α-reductase that normally suppresses testosterone conversion into 5α-dihydrotestosterone, and thus, arrests the proliferation of prostate cancer cells [62]. Another recent work showed that OA strongly suppresses the proliferation of esophageal cancer cells (OE19, OE33), most likely by activating suppressor genes p53, p21 and p27 [63]. In breast cancer cells SK-Br3 and BT-474, OA suppressed the expression of human epidermal growth factor receptor (*Her-2/neu*) oncogene that plays an important role in the progression of certain malignancies and ovary, stomach and uterus cancer [10,64,65]. The proapoptotic effect of OA was observed when it enhanced the production of reactive oxygen species and increased the activity of caspase-3 in lymphoma YAC-1 cells [66]. OA administration to mice carrying lung adenocarcinoma LAC-1 resulted in a significant retardation of tumor growth and increased longevity of the animals [67]. Hence, a large number of studies carried out so far, in vitro and in vivo, evidence that OA displays cytotoxicity towards a variety of cell types and can suppress tumors utilizing different mechanisms.

## 2. Materials and Methods

### 2.1. Reagents

The following reagents were used. Aniline naphthalene-8-sulfonic acid (1,8-ANS), dimethylsulfoxide (DMSO) and fatty acids, including oleic acid (OA), linoleic, arachidonic, linolenic, palmitic and stearic acids, were from Sigma-Aldrich (Saint Louis, MO, USA); NaCl, Na_2_HPO_4_, NaH_2_PO_4_ and HCl were from Merck KGaA (Darmstadt, Germany); ethanol, methylene blue and formalin were from Serva (Heidelberg, Germany); Bio-Gel P-6 was from Bio-Rad (Hercules, CA, USA).

In regard to recombinant lactoferrin, the milk of transgenic goats was used to isolate recombinant human lactoferrin (rhLF) as described previously [68]. Briefly, milk was centrifuged at 20,000× *g* for 30 min at 4 °C to remove the fat and then was acidified to pH 4.6 by adding 1 M HCl at 37 °C for 30 min to precipitate casein. The precipitate was removed by centrifugation (20,000× *g* for 30 min), the pH of the supernatant was adjusted to 7.0 by adding 1 M NaOH and the second step of centrifugation was performed to collect the final rhLF-enriched supernatant. The supernatant was loaded onto a TOYOPEARL SP-550 cation-exchange column (Tosoh Bioscience, Griesheim, Germany), which was equilibrated and washed with a buffer containing 20 mM sodium phosphate and 0.4 M NaCl at pH 7.0. A linear gradient of 0.4–1.0 M NaCl was used to elute cationic proteins, and the fractions with rhLF were pooled. After dialysis/filtration, the content was concentrated by ultrafiltration using Sartorius Vivaflow-200 crossflow (Sartorius, Göttingen, Germania) cassette with 30 kDa MWCO membrane, and lyophilized. The purity of the rhLF was verified by SDS polyacrylamide gel electrophoresis, which showed a single band of ~80 kDa. The percentage of rhLF purity was not less than 99%, with iron saturation not higher than 10%. LF eluted from the TOYOPEARL SP-550 column was subjected to an additional cation-exchange chromatography on a Mono S column (Amersham Pharmacia Biotech, Uppsala, Sweden) to separate the rhLF from goat LF. The concentration of rhLF in the milk of transgenic goats ranged from 2 g/L to 16 g/L, which significantly exceeded the content of goat LF in the milk of various caprine breeds (0.073–0.089 g/L).

### 2.2. Experimental Animals

Male C3HA mice (16–26 g) were obtained from Rappolovo farm (Rappolovo, Leningrad Region, Russia). Animals were kept under regular vivarium conditions receiving standardized forage and water.

### 2.3. Cell Cultures

The hepatoma 22a cell culture was obtained from the Biobank at the Institute of Cytology of the Russian Academy of Sciences (1-20/27.02.2020, Saint-Petersburg, Russia). Hepatoma 22a had been induced in 1951 in C3HA mice using ortho-aminoazotoluene, after which it was adapted to in vitro culturing. Cells were cultured in DMEM (Biolot, Russia) supplemented with 10% fetal calf serum (FCS; HyClone, Cramlington, UK), 0.1 mg/mL gentamycin (Biolot, Moscow, Russia) and 0.6 mg/mL glutamine (Biolot, Moscow, Russia) at 5% CO_2_ and 37 °C. The dense monolayer of hepatoma 22a cells was washed with a physiological solution, after which the cells were detached by keeping them for a minute in a trypsin:Versene solution (1:1) at 37 °C. They were resuspended in DMEM containing 0.6 mg/mL glutamine, 0.1 mg/mL gentamycin and 5% FCS and transferred into Carrel’s flasks.

### 2.4. Obtaining Complexes of LF with Fatty Acids. Assessing LF/FA Stoichiometry

Next, 4 mL of LF (120 mg/mL) in phosphate buffer saline (PBS) were used to prepare complexes. Then, 1 mL of PBS was layered above that solution, and 100 µL of ethanol was cautiously dropped. Fatty acid (FA) was dissolved in a small volume of ethanol and mixed. To obtain complexes, the solution of LF was titrated with aliquots of FA (from 2:1 to 8:1 molar excess). Upon adding another portion of FA, the solution was thoroughly mixed on a vortex (2400 r.p.m.) at room temperature. Three mixings were done after every addition at 1 min intervals. Excessive FA was eliminated by overnight dialysis against PBS at 4 °C followed by passing the solution through 0.45 µM filter. The protein concentration after the dialysis was assayed at 280 nm on a spectrophotometer SF-2000-02 (OKB-Spectr, Saint Petersburg, Russia). The amount of FA bound to LF was assayed by colorimetry at 550 nm using the enzymatic kit NEFA (Randox, Crumlin, UK).

A column (100 × 5 mm) packed with Bio-Gel P-6 (Bio-Rad, Hercules, CA, USA) was used for gel filtration of the LF-8OA complex at a flow speed of 0.25 mL/min. Then, 100 µL of LF-8OA (40 mg/mL) were loaded on the column equilibrated with PBS. The column was washed with PBS and the effluent was collected in 500 µL fractions. Protein in the fractions was detected by absorption at 280 nm. The enzymatic kit NEFA (Randox, Crumlin, UK) was used to measure FA in a fraction with maximum protein concentration. The relation of the protein to the OA content in the sample prior to gel filtration was assessed and compared with that of the LF-containing fraction eluted from the column.

### 2.5. Fluorescence and Circular Dichroism Spectra

#### 2.5.1. Registering Circular Dichroism Spectra

To register the circular dichroism (CD) spectra protein samples were dissolved in a phosphate-saline buffer (0.15 M NaCl, 10 mM sodium-phosphate, pH 7.4) to obtain concentrations of 0.1 mg/mL (far UV) and 1 mg/mL (near UV), using the spectropolarimeter Jasco J-810 (Jasco, Tokyo, Japan). Samples were kept at 25 °C for 10 min before spectroscopy. Using a quartz cuvette with an optical path of 1 mm, far UV spectra were registered in the range of 190–250 nm at a scanning pitch of 0.1 nm (spectrometer slit was 2 nm). For near UV spectra, a cuvette with an optical path of 10 mm was used and the spectra were registered in the range of 250–360 nm, other parameters being the same. To avoid the increment of low-frequency random processes, CD spectra for each sample were registered four times and then averaged. Smoothing of the spectra was achieved against the CD spectra obtained for PBS using software provided by the manufacturer of the Jasco J-810 (Jasco, Tokyo, Japan). The results of three separate experiments were averaged. Data obtained were expressed as molar ellipticity ([θ]), deduced from the equation [69]:θ = (MRW × θ)/(10 × c × l),(1)
where θ is ellipticity (°), c is protein concentration (g/mL), l is the optical path (cm) and MRW was calculated according to the formula:MRW = MM/(Na.r. − 1),(2)
where MM is the molecular mass of the protein (g/mol) and Na.r. is the amount of the amino acid residues in the polypeptide.

#### 2.5.2. Registering Fluorescence Spectra

Using a quartz cuvette, fluorescence spectra were registered in the range of 305–500 nm on a Jasco EP-6500 fluorimeter (Jasco, Tokyo, Japan), at excitation wavelengths of 280 nm and 295 nm. Samples were kept at 25 °C for 10 min before spectroscopy. Results obtained from three measurements were averaged.

To obtain the complexes LF/ANS and LF-8OA/ANS, each protein sample was incubated for 1 h with a fivefold excess of ANS at room temperature in the dark. Emission spectra were registered in the range of 400–700 nm, at an excitation wavelength of 390 nm.

### 2.6. Studying Hepatoma 22a Viability and Lysis of Erythrocytes Induced by LF and LF-8OA

#### 2.6.1. Viability of Hepatoma 22a Cells

The direct cytotoxic effect on tumor cells in vitro was assessed using 24-well plates with 5 × 10^4^ hepatoma 22a cells in 500 µL DMEM with 5% FCS per well. Cells were cultured until a subconfluent monolayer was formed. The samples under testing were added at concentrations varying from 3.25 µM to 25 µM (LF, LF-8OA, 8OA) and incubated for 24 h at 37 °C. A stock solution of 20 mM OA in DMSO was added to the medium and cells so that the final concentration of DMSO did not exceed 1%. After that, the cells were fixed in 10% formalin and stained with 0.05% methylene blue. The dye was dissolved in 0.3 M HCl and the absorption was measured at 620 nm. The viability of cells was calculated according to the formula:V = As × 100%/Ac,(3)
where As is absorption in the wells with samples under study and Ac is absorption in the wells with hepatoma cells to which the phosphate-saline buffer was added instead of the samples under study.

#### 2.6.2. Isolation of Erythrocytes from Peripheral Blood and Studying the Hemolysis

Erythrocytes were isolated from the citrate-treated blood of healthy donors after triple washing with PBS to eliminate plasma, followed by centrifugation at 600× *g* for 10 min. The erythrocytic mass obtained was diluted with PBS to the ratio cells:buffer = 1:8, and incubated in 96-well plates for 1 h at 37 °C, with the samples under study diluted to the concentration needed in a thermal shaker (290 r.p.m.).

Hemolysis was assessed in a plate spectrophotometer CLARIOstar (BMG Labtech, Ortenberg, Germany) by absorption at 412 nm (Soret band) in the supernatant, transferred to a clean 96-well plate after precipitating the erythrocytes. The intensity of hemolysis was calculated using the equation:H = 100% × (ODs − ODc)/(OD100 − ODc),(4)
where ODs is the optical density of the supernatant in the samples under study, ODc is the optical density in the control wells and OD_100_ is the optical density of the fully lyzed cells.

The control wells contained a sodium-phosphate buffer, which was substituted for pyrogen-free deionized water (Mediana-Filter, Moscow, Russia) when complete cell lysis was required. The results of the two separate experiments, with four repeated measurements in each, were averaged.

### 2.7. Studying the Effect of LF and LF-8OA on Mortality of Mice with Hepatoma 22a

Hepatoma 22a cells were detached as described above, resuspended in physiological solution (DPBS) and serum proteins were washed off by centrifugation at 800× *g* for 10 min. Washing was repeated thrice. Live and dead cells stained with 3% Trypan blue were calculated in a cell chamber. Hepatoma 22a cells (2 × 10^5^ in 0.25 mL of the DPBS per animal) were inoculated subcutaneously into the dorsal region of each mouse (n = 30). All animals were divided in three experimental groups. The control group received the physiological solution, another group received recombinant human LF (5 mg per mouse) and animals in the third group were injected with the same dose of LF-8OA complex. The dorsal subcutaneous injections of the preparations to mice of the second and third groups started on the day after the tumor inoculation. Such daily injections were continued for 24 days. The moment of a tumor’s appearance, its size and the time of the animal’s death were registered every second day. Beginning from the 10th day past the tumor inoculation, the size of the malignant node was measured in all mice and its volume was determined using the formula:V = (a × b^2^)/2,(5)
where a and b are the length and width of the tumor node, respectively.

The antitumor activity of the complex formed by LF and OA was assessed as tumor growth retardation (TGR) and tumor growth index (TGI) expressed in percentages. The TGR percent was determined using the formula:TGR = 100% × (Vc − Ve)/Vc,(6)
where Vc and Ve are the mean volume of the tumor (mm^3^) in the control and experimental group, respectively.

The TGI was determined using the formula:TGI = Se/Sc,(7)
where Se is the area (mm^2^) under the kinetic curve of tumor growth in mice receiving substances under study and Sc is the area (mm^2^) under the kinetic curve of tumor growth in mice of the control group.

To measure the area under the kinetic curve of tumor growth, the trapezium method was used, represented by the following equation:S = t_1_ × (V_1_ + V_2_)/2 + t_2_ × (V_2_ + V_3_)/2 + … + t_n−1_ × (V_n−1_ + V_n_)/2,(8)
where: V_i_ is the volume (mm^3^) of a tumor numbered I, n is the number of measurements, t_1_ is the time between the first and the second measurement (days), t_2_ is the time between the second and the third measurement (days) and t_n−1_ is the time between the ultimate and the penultimate measurement (days).

The study involving mice was approved by the Ethical Committee of the Institute of Experimental Medicine, protocol No 7-09RA/2020 stating the compliance with the ARRIVE 2.0 guidelines (14 July 2020).

## 3. Results

### 3.1. Specificity of LF Interaction with Fatty Acids

When studying the interaction of LF with a number of non-saturated (arachidonic, linoleic, linolenic and oleic) FA dissolved in ethanol, we observed no micelle formation typical of mixing FA with a physiological solution. Protein titration showed that 1 mole of LF is capable of binding up to 8 moles of OA, or nearly 6 moles of linoleic acid, and almost 5 moles of linolenic and arachidonic acids (Table 1). Saturated FA (palmitic, stearic) did not interact with LF. No traces of FAs were detected in the initial LF preparation. 

The results of measuring the protein-bound OA before and after the gel filtration of the LF-8OA complex on a Bio-Gel P-6 column are summarized in Table 2. Since 1 mole of LF bound 7.8 ± 0.2 and 7.6 ± 0.2 moles of FA before and after gel filtration, repectively, it can be concluded that virtually all OA was within the complex with LF and the presence of unbound OA in a solution can be ruled out.

### 3.2. Tryptophan Fluorescence and CD Spectra of LF and LF-8OA

The intrinsic fluorescence of LF and of its complex with OA was registered. Excitation wavelengths were specified corresponding to the absorption maximum for tyrosine plus tryptophan (total fluorescence λ_exc_ = 280 nm) and that for tryptophan alone (λ_ex_ = 295 nm). Figure 1a,b shows the intrinsic fluorescence spectra of tryptophan in the preparations studied. It is seen that excitation at 280 and 295 nm resulted in emission with λ_max_ = 330–330.2 nm, which is evidence of substantial contribution of Trp into the fluorescence spectra.

Neither in LF itself, nor in LF-8OA, were any significant differences between the intensity of the total fluorescence (Tyr + Trp) and that of Trp alone were observed. Indeed, the preparations studied had similar spectra with the maximum at 330 nm, which rejects the assumption of substantial structural perturbations in LF upon adding OA in eightfold molar excess.

Figure 2 shows the fluorescence spectra of LF and LF-8OA upon binding the hydrophobic probe 1-aniline naphthalene-8-sulfonic acid (1,8-ANS).

Free ANS showed practically no light emission in the sodium-phosphate buffer. In contrast, the fluorescence intensity of LF and of its complex with OA became increased dozens of times, which is explained by the capacity of ANS to interact with clusters of hydrophobic amino acids in the protein.

Binding 1,8-ANS with LF and with LF-8OA shifted the fluorescence maximum to ~479 nm and 476 nm, respectively. The fluorescence intensity of LF in complex with OA was higher than the same feature of LF alone, which may be due to a larger number of hydrophobic sites available for ANS in the complex LF-8OA.

The effect of OA on the secondary structure of LF in the phosphate-saline buffer was studied by registering CD spectra in the far UV region (200–250 nm). No significant differences were observed between the spectrum of LF and of its complex with OA. Both spectra had noticeable negative ellipticity in the region of 209–224 nm, which is typical for an α-helix structure. The same conformity of CD spectra was registered for LF, and for LF-8OA, it was observed in the near UV region (Figure 3). It can be concluded that adding the amounts of OA used in our experiments to LF did not alter its secondary structure.

### 3.3. Effect of LF and LF-8OA on Hepatoma 22a Cells and on Stability of Erythrocytes

Cytotoxic activity of LF and LF-8OA was assessed in vitro using cultured normal and malignant cells. The direct cytotoxic effect of LF complexed with OA on hepatoma 22a cells was evident. Indeed, the complex displayed dose-dependent antitumor activity. as increasing its concentration from 3 to 25 µM diminished the percentage of viable cells in comparison with the control culture (Figure 4). Neither LF without OA, nor OA prepared as stock solution in DMSO, had such a pronounced effect on the viability of hepatoma 22a cells.

It is noteworthy that the viability of hepatoma H22a cells decreased by 16% upon treatment with 12.5 µM LF-8OA, while the same concentration of LF alone or 100 µM OA (8 × 12.5 µM) did not affect their viability. Taken separately at higher concentrations, i.e., 25 µM LF and 200 µM OA (8 × 25 µM), these two factors caused the viability to decrease by 11% and 13%, respectively. Meanwhile, it decreased by 34% when cells were treated with 25 µM LF-8OA. Hence, the effect of the complex LF-8OA on the viability of tumor cells cannot be explained by a summation of the effects produced by LF and OA.

Adding LF-8OA at a concentration of 8 µM, and more to freshly isolated human erythrocytes, caused increasingly noticeable hemolysis, while LF alone had no hemolytic activity (Figure 5).

### 3.4. Effect of LF and LF-8OA on Mortality of Mice with Hepatoma 22a

The changes of tumor volume in mice were continuously registered until the moment of an animal’s death. Kinetic curves were plotted to monitor the dynamics of hepatoma 22a growth (Figure 6). Rapid growth was observed in the control group. Meanwhile, a single daily injection of LF-8OA (5 mg per mouse) caused growth retardation registered at every stage of the monitoring. In some cases, it resulted in a complete resorption of the tumor node. LF injected into mice without OA did not cause such a tumor-inhibitory effect.

On day 30 after the tumor inoculation, the TGR in mice that received LF-8OA was about 79% as compared with the control group. The same index was 21% in mice treated with LF alone (Figure 7).

Careful calculations of the TGI, that presents the efficiency of antitumor therapy, yielded 0.14 for LF-8OA and 0.63 for LF.

The retardation of tumor growth by the complex LF-8OA resulted in increased longevity of mice, as compared with the control group and the animals that received only LF. In the control group, the first mice died on day 30 after the tumor inoculation. In the group receiving only LF, the first death was registered on day 15 after the tumor inoculation. At that moment, the survival in the group treated with LF-8OA was 100%.

On day 48, all mice died in the control group and in the group that received only LF. Meanwhile the survival among the animals treated with LF-8OA was 70%. On days 73 and 106, this index was 50 and 20%, respectively. In 20% of the animals, the tumor nodes were fully resolved. Figure 8 shows the plot illustrating the survival in all groups of tested animals. Comparing the three survival curves in Figure 8 shows that they differ with *p* = 0.016 (χ^2^ = 13.705). Gehan’s Wilcoxon Test was applied to compare the survival curves obtained for the control and LF-8OA groups, and showed significant differences between them (*p* = 0.00271). The same test also indicated significant differences between the curves obtained for the LF and LF-8OA groups (*p* = 0.00167).

## 4. Discussion

As specified in the Introduction, the starting point for this work was a report on the capacity of cow-milk LF in complex with OA to suppress the growth of tumor cells [9]. One of the aims of this study was to demonstrate the relative specificity of the interaction between LF and OA, but also to show that human LF in complex with OA is an efficient inhibitor of tumor growth.

Among the non-saturated fatty acids tried in our experiments properly, OA showed the highest binding with LF. Importantly, no saturated fatty acid got bound to that protein (Table 1).

We showed previously that LF-8OA complex forms a monodisperse system with a gyration radius of 8 nm, which is almost two times bigger than the same feature of LF, i.e., of 4.8 nm [58].

Moreover, in view of the fact that cow-milk LF in complex with OA had altered tryptophan fluorescence as well as CD spectrum [9], we checked whether similar changes occur in human LF upon its interaction with OA. Interestingly, neither intrinsic fluorescence nor secondary structure elements in human LF underwent considerable alterations (Figure 2 and Figure 3). However, ANS, the hydrophobic probe, showed a noticeable increase of hydrophobicity in LF upon its interaction with OA. A cautious suggestion can be made that gradual titration of the protein with fatty acid in the presence of a minimum of ethanol (later eliminated by dialysis) allowed us to obtain the barely denatured complex LF-8OA.

Cytotoxicity of that complex was tested on erythrocytes, which serve as the simplest model for testing the agents disturbing the stability of the cell membrane. It was the LF-8OA complex that caused hemolysis in our experiments, unlike LF without a fatty acid (Figure 5). It seems likely that LF-8OA is capable of forming pores in cell membranes. On account that such complex may be toxic for non-malignant cells, we performed subcutaneous, but not intraperitoneal, injections to mice used in experiments with tumor growth.

In the course of treatment with LF-8OA (days 1–24 after the tumor inoculation) not a single mouse died in that group, so, a conclusion can be made about the absence of acute toxicity of the complex in the dose used. Moreover, only in that group did quite a few animals remain alive seven weeks after the start of the experiment. Thus, in contrast to the preparation of comparison (LF), the LF-8OA complex caused a significant increase of the animals’ survival rate and a complete resolution of tumor nodes in 2 animals out of 10 in the group (Figure 8).

We chose OA on account of the fact that among fatty acids, it has the highest capacity to get bound by LF. Besides, a large bulk of evidence concerning its antineoplastic activity can be found in the literature. Recently, the antitumor effect of LF and linolenic acid was studied on a murine model [70].

Considering the lipolytic effect of LF and the prevalence of OA among fatty acids in mammalian tissues, it cannot be excluded that LF and OA can form a complex in vivo. Testing this hypothesis requires another series of experiments.

## 5. Conclusions

The results obtained clearly show the capacity of LF in combination with OA to inhibit the growth of hepatoma 22a, chosen as a model in our experiments. At least two important observations were made, i.e., the presence of a dose-dependent antitumor effect of the complex LF-8OA, and the significant suppression of tumor growth in vivo caused by that complex, which resulted in the increased longevity of the tumor-carrying animals. It is worth noting that in 20% of mice with hepatoma, the tumor nodes disappeared in the course of treatment with LF-8OA. Since LF alone had none of these effects, but on the contrary stimulated tumor progression, it can be concluded that OA imparts antitumor activity to its complex with LF. In this study, we did not explore the molecular mechanisms of the antitumor effect of the complex formed by LF and oleic acid. Details of such mechanisms will be the aim of a future investigation.

## Figures and Tables

**Figure 1 materials-14-01602-f001:**
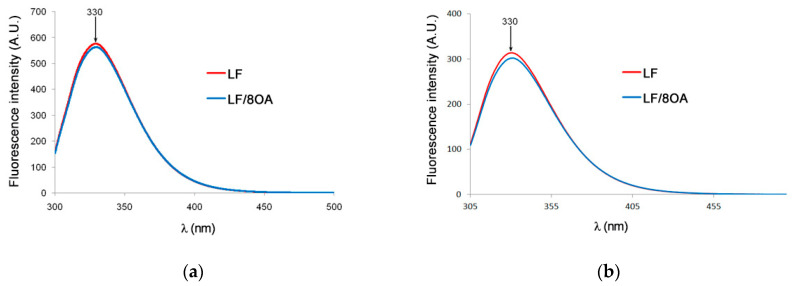
Intensity of intrinsic fluorescence of LF (red line) and LF-8OA (blue line) in phosphate-saline buffer (0.15 M NaCl, 10 mM sodium-phosphate buffer, pH 7.4): (**a**) λ_exc_ = 280 nm; (**b**) λ_exc_ = 295 nm. Protein concentration was 1 mg/mL. Spectra were registered at 25 °C on a JASCO EP-6500 spectrophotometer. Results show the mean value ± SD (n = 3).

**Figure 2 materials-14-01602-f002:**
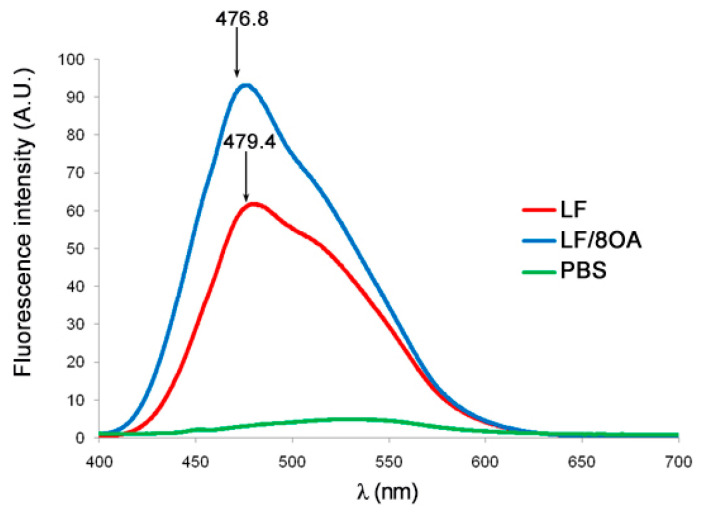
Intensity of fluorescence of LF (red line) and LF-8OA (blue line) in phosphate-saline buffer (0.15 M NaCl, 10 mM sodium-phosphate buffer, pH 7.4) upon binding at room temperature to a fivefold excess of 1-aniline naphthalene-8-sulfonic acid (1,8-ANS) for an hour at dark. The protein concentration was 1 mg/mL. The excitation wavelength was 390 nm. Spectra were registered in the range of 400–700 nm. Results show the mean value ± SD (n = 3).

**Figure 3 materials-14-01602-f003:**
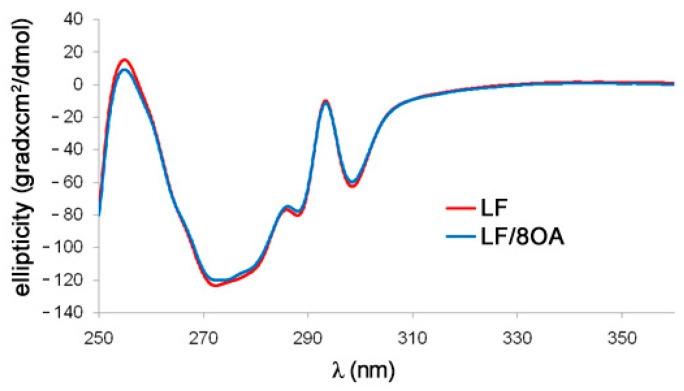
Circular dichroism (CD) spectra of LF and LF-8OA in the near UV region. Spectra were registered at room temperature on a JASCO J-810 spectropolarimeter in the range of 250–360 nm with 0.1 nm pitch (spectral slit width 2 nm), in a phosphate-saline buffer (0.15 M NaCl, 10 mM sodium-phosphate buffer, pH 7.4). Solid red and blue lines are the spectra of LF and LF-8OA, respectively. Protein concentration was 1 mg/mL. Three independent experiments were done in which each spectrum was averaged over four scannings. CD spectra of the buffer solution served as baseline.

**Figure 4 materials-14-01602-f004:**
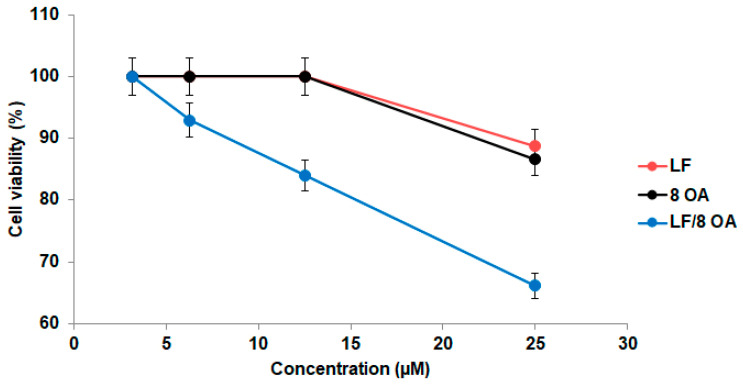
Viability of H22a cells determined by methylene blue assay upon 24 h of treatment with 3.13, 6.25, 12.5 and 25 µM LF (red circles) or LF-8OA (blue circles), and with 25, 50, 100 and 200 µM OA (black circles), which corresponds to 8 moles of OA in its complex with LF, taken in the range of concentrations 3–25 µM. Results show the mean value ± SD (n = 3).

**Figure 5 materials-14-01602-f005:**
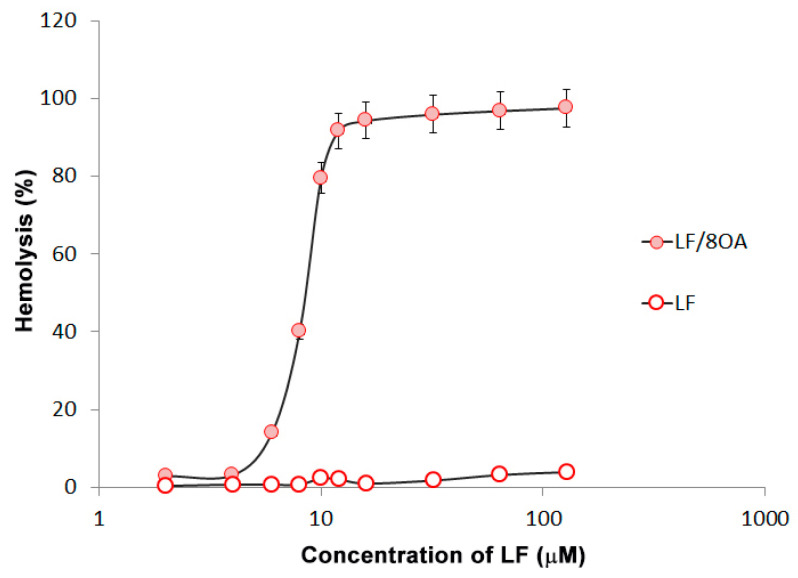
Concentration-dependent hemolytic effect of LF-8OA in comparison with LF. Efficiency of hemolysis was assessed upon 1 h of human erythrocytes incubation with LF (white circles) and LF-8OA (red circles) at 37 °C. In control experiments, a phosphate-saline buffer (0.15 M NaCl, 10 mM sodium-phosphate buffer, pH 7.4) was used, which was substituted for deionized water to obtain complete hemolysis. Results show the mean value ± SD (n = 3).

**Figure 6 materials-14-01602-f006:**
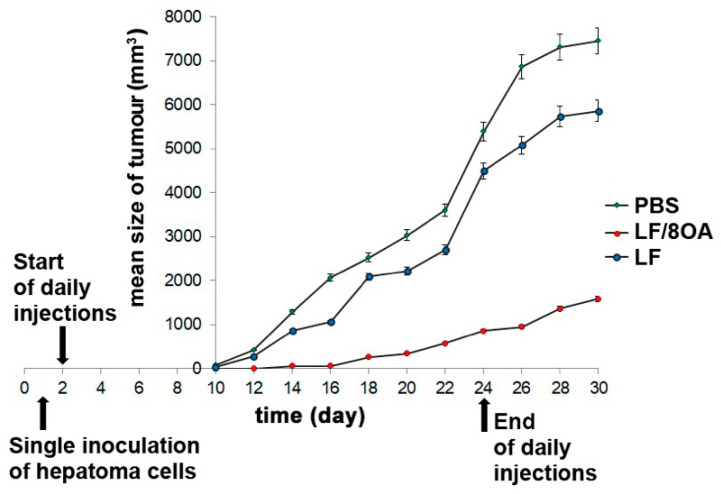
Tumor growth dynamics in C3HA mice after inoculation of 2 × 10^5^ hepatoma 22a cells and administration of the preparations under study. Plots represent mean tumor volumes in mice from the control group (green circles), LF-treated group (blue circles) and LF-8OA-treated group (red circles). Daily injections of preparations (5 mg per mouse) began on the day after the tumor inoculation and lasted for 24 days. Starting from the 10th day, tumor size was measured in every mouse.

**Figure 7 materials-14-01602-f007:**
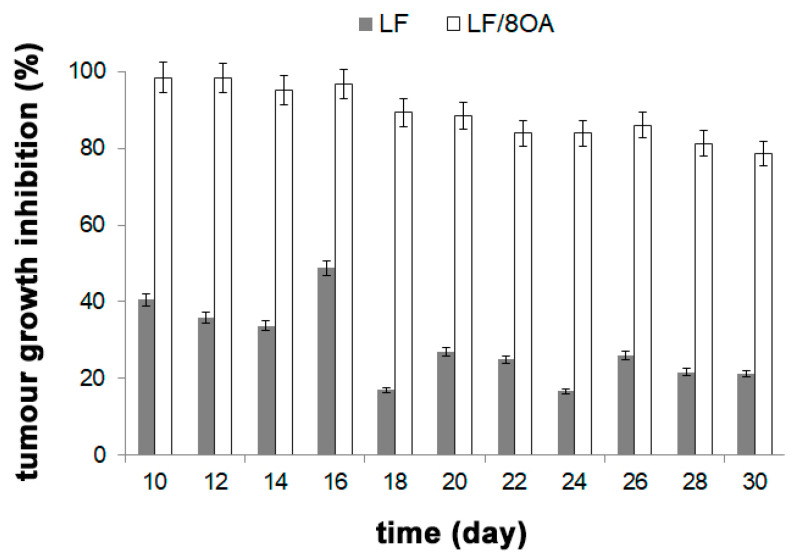
Dynamics of hepatoma 22a growth retardation in C3HA mice after administration of LF or LF-8OA. Grey and white columns represent the process of hepatoma growth retardation in LF-treated and in LF-8OA-treated mice, respectively.

**Figure 8 materials-14-01602-f008:**
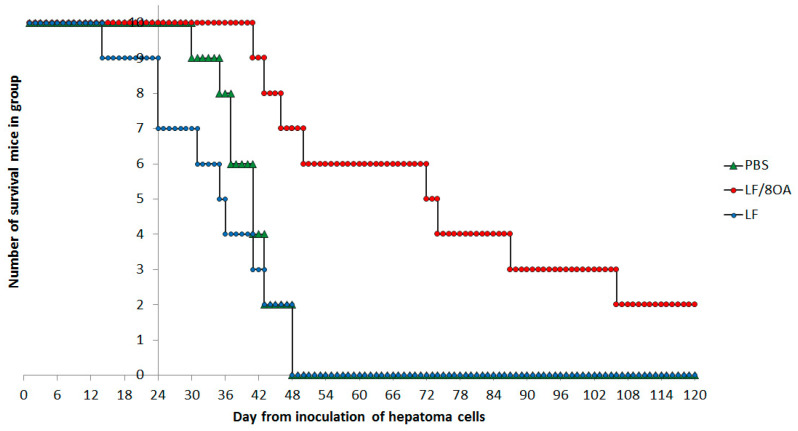
Kaplan–Meier curves of survival rate within 24 days after the H22a cells inoculation for PBS- (green), LF- (blue) and LF-8OA-treated (red) mice. Each group consisted of 10 animals. Every point on a curve means the number of mice that remained alive by the day of observation.

**Table 1 materials-14-01602-t001:** Stoichiometry of interaction between lactoferrin (LF) and fatty acid (FA).

Fatty Acid	FA/LF (Mole/Mole)
Linolenic	4.7
Arachidonic	5.5
Linoleic	5.8
Oleic	8.4

**Table 2 materials-14-01602-t002:** The amount of LF and the FA/LF relation in a sample of LF-8OA before and after gel filtration on a Bio-Gel P-6 column.

	LF (mg)	FA/LF (Mole/Mole)
Before loading	4.0 ± 0.2	7.8 ± 0.2
In eluted fraction	3.8 ± 0.2	7.6 ± 0.2

## Data Availability

The data presented in this study are available on request from the corresponding author.

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
