# Peer review of "Interaction of Lactoferrin with Unsaturated Fatty Acids: In Vitro and In Vivo Study of Human Lactoferrin/Oleic Acid Complex Cytotoxicity"

_materials, 2021, doi:10.3390/ma14071602_

Round 1

Reviewer 1 Report

  1. Please introduce why the percentage (1:8) of LF/OA was used in this study in the “introduction “section. On the other hand, whether the percentage is efficient for other cancer types?
  2. At line 62 and 70, please indicated which tumor types were studies in previous studies
  3. At table I, authors showed the LF is cable of binding to various non-saturated FAs such as arachidonic, linoleic, linolenic and oleic FA. Whether only FA/OA has the anticancer effects on hepatoma 22a? Please showed the cytotoxic index of hepatoma 22a with LF/5.5arachidonic FA, LF/5.8 linoleic FA and LF/4.7 linolenic FA.

Author Response

  1. Please introduce why the percentage (1:8) of LF/OA was used in this study in the “introduction “section. On the other hand, whether the percentage is efficient for other cancer types?

RESPONSE: Our research was focused on studying properly the malignant tumour known as hepatoma. We made preliminary tests of LF/8OA complex effect on such cell lines as HL-60, THP-1, Jurkat and HeLa, and finally chose H22a, since it is very convenient for experimentation in mice. Upon scrutinizing the “Introduction” we did not find an appropriate place for adding an explanation why “the percentage (1:8) of LF/OA” was used. An appropriate explanation is presented in the “Results”.

2. At line 62 and 70, please indicated which tumor types were studies in previous studies

RESPONSE: The phrase in line 70 contains the reference [9] of the paper by Fang et al., in which tumour type is specified.

3. At table I, authors showed the LF is cable of binding to various non-saturated FAs such as arachidonic, linoleic, linolenic and oleic FA. Whether only FA/OA has the anticancer effects on hepatoma 22a? Please showed the cytotoxic index of hepatoma 22a with LF/5.5arachidonic FA, LF/5.8 linoleic FA and LF/4.7 linolenic FA.

RESPONSE: As explained in “Introduction”, our research was rooted in the report by Fang et al., 2014 (doi: 10.1016/j.bbalip.2013.12.008) who showed the prominent capacity of the complex formed by oleic acid and lactoferrin to suppress tumour growth. In view that other fatty acids tested in our study bound to LF at lower molar ratios, those were not of much interest to us and their anticancer effect was not explored. A paper by Li et al., 2020 (doi: 10.1021/acsomega.0c01132) comparing effects of oleic and linoleic acids in the presence of LF is cited in our manuscript (Ref. 70). Certainly, we did not test a complex FA/OA as suggested by the Reviewer, since we do not see any sense in combining two fatty acids (besides, it is unclear if those can ever make a complex as suggested).

Reviewer 2 Report

Journal: MDPI - Materials

Manuscript

Title:  " Interaction of lactoferrin with unsaturated fatty acids: in vitro and in vivo study of human lactoferrin/oleic acid complex cytotoxicity"

Author(s): Anna Elizarova, Alexey Sokolov, Valeria Kostevich, Ekaterina Kisseleva, Evgeny Zelenskiy, Oleg Panasenko, Alexander Budevich, Igor Semak, Vladimir Egorov, Giulia Pontarollo, Vincenzo De Filippis and Vadim Vasilyev

Reviewer Comments to Author(s)

Recommendation: Major revisions

In this articles the authors researched on the in vivo evaluation of oleic acid in complex with lactoferrin. This is an interesting research report, however there are major concerns. Almost 44 out of 70 references are of the past decade (before 2010). This is almost 60 % of the references. The author(s) maybe should consider of using more up to date references to support their findings in a state-of-the-art manner.

The author(s) might consider the following:

  1. The abstract contains sufficient information but their presentation could be corrected, so that they have a logical order in their presentation. The author(s) could first present information about their system and its characterization and then the data for cellular studies and finally for animal studies. For example, line 27-28 could be moved below as it reports information presented at the final part of abstract.
  2. Introduction and the whole manuscript is lacking of a state-of-the-art. In lines 44 – 60 the paragraphs are confusing. Firstly, Surgery, radiation treatment and chemotherapy represent conventional therapeutic approaches. However, novel techniques and drug treatments based on drug delivery systems, such as liposomes (a basic example is Doxil) represent state-of-the-art therapeutic approaches. The author(s) are kindly advised to provide more specifications and up-to-date references for this state-of-the-art paragraph they use. Moreover, next paragraph lacks of sufficient evidence. The critical points that support the application of low toxic compounds are presented superficially and there are significant shortcomings in the documentation of the authors' allegations. Which are the "novel compounds"? Could the author(s) present some examples of compounds and applications related to their research? Where are the references? Where are the up-to-date examples of applications? In line 57 the author(s) use the expression "...arrest their propagation." The propagation or the diffusion of tumor cells cannot be arrested. Can be limited or reduced or something similar. However, the author(s) should provide references supporting their statements and explanations on the meaning of the phrase. Could the author(s) please explain are there natural molecules (of which type) that can restrict tumor cells' diffusion? Example should be provided. Moreover, could the author(s) specify by "propagation" do they refer to tumor cells' diffusion or proliferation?
  3. In line 73 what do the author(s) mean by “… investigation of this phenomenon is really topical”?
  4. Line 101-102. This sentence is confusing and should be rephrased by the author(s).
  5. The author(s)are kindly requested to use the correct indicators for the numbers in line 160.
  6. What is the point of parts 2.2 and 2.3? The author(s) could use one part for in vitro studies presenting all their protocols and procedures in detail and another one for in vivo animal studies. This kind of presentation the author(s) use is confusing.
  7. Line 225, Concentrations, controls and samples should be mentioned in detailed by the author(s).
  8. Line 248, Could the author(s) explain why they used this specific concentration of LF? Is it referenced or has been studied elsewhere? Which was the injection plan the author(s) followed? Was only a single injection of 5 mg, was it repeated injections of 5mg or repeated injections of a total of 5 mg? The injection plan should be provided.
  9. The control culture is not presented and not analyzed, which is the control culture that author(s) used? results of the normal cell evaluation are also not presented by the author(s). Moreover, the results of LF alone are not presented at all. The author(s) are kindly requested to present all of their results that they refer to. In line 322 the author(s) state that the LF-8Oa system caused "increasingly noticeable hemolysis". Are they sure of their result? Hemolysis is the rupture or destruction of red blood cells, and the author(s) state that the system they used in animal studies can cause increasingly noticeable hemolysis? The author(s) are kindly requested to present specifications on the red blood cells rupture by the LF-8OA.
  10. Could the author(s) explain whether the animals were sacrificed at the end of the study and the tumors were collected or not? If yes could they please add photos of the tumors and dimensions for comparison reasons?
  11. Line 358, The morphology of the LF-8OA system is unclear. Could the author(s) provide more proofs and specifications on this subject?
  12. Line370-371 “It seems likely that LF-8OA is capable of forming pores in cell membrane “. How did the authors come to this conclusion?

Author Response

In this articles the authors researched on the in vivo evaluation of oleic acid in complex with lactoferrin. This is an interesting research report, however there are major concerns. Almost 44 out of 70 references are of the past decade (before 2010). This is almost 60 % of the references. The author(s) maybe should consider of using more up to date references to support their findings in a state-of-the-art manner.

RESPONSE: The authors are grateful to the Reviewer for having read the manuscript so attentively. It is not clear what means “Almost 44 out of 70 references…”. Which is that number? Is it 43.5 or e.g. 43.75 that makes it nonintegral?  Besides, it is not clear if the Reviewer would like to see more recent works cited only to support our findings or he/she is not satisfied by the references of the past decade that introduce a reader in the history of the problem. Antitumour properties of LF have been studied since the 1990-ies and about that time HAMLET was first described, hence, it seems reasonable to present an overview of the problem without too many details. Not to go in discussions we added some references as suggested.

  1. The abstract contains sufficient information but their presentation could be corrected, so that they have a logical order in their presentation. The author(s) could first present information about their system and its characterization and then the data for cellular studies and finally for animal studies. For example, line 27-28 could be moved below as it reports information presented at the final part of abstract.

RESPONSE: We fully agree with the Reviewer and have made the necessary amendments in the Abstract.

2. A) Introduction and the whole manuscript is lacking of a state-of-the-art. In lines 44 – 60 the paragraphs are confusing. Firstly, Surgery, radiation treatment and chemotherapy represent conventional therapeutic approaches. However, novel techniques and drug treatments based on drug delivery systems, such as liposomes (a basic example is Doxil) represent state-of-the-art therapeutic approaches. The author(s) are kindly advised to provide more specifications and up-to-date references for this state-of-the-art paragraph they use.

RESPONSE: The authors are aware of the recent developments in treatment of malignancies, but do not understand what is wrong about considering e.g. radiation treatment and chemotherapy the state-of-the-art techniques. Ion therapy (proton, carbon, etc.) so far is introduced only in several most advanced medical centres, and it is surely one of such. The emerging techniques involving target therapy or CAR-T applied when substitution of malignant cells in bone marrow with healthy ones might be classified as chemotherapy until another term becomes stabilized. These are only two examples of novel therapeutic approaches that are not likely to be termed as “conventional”, while surgery is not a therapeutic approach at all. Our paper is dedicated to the results of a particular experimental study rather than to reviewing all existing modes of medical interventions.

B) Moreover, next paragraph lacks of sufficient evidence. The critical points that support the application of low toxic compounds are presented superficially and there are significant shortcomings in the documentation of the authors' allegations. Which are the "novel compounds"? Could the author(s) present some examples of compounds and applications related to their research? Where are the references? Where are the up-to-date examples of applications?

RESPONSE: It is not understood what kind of evidence and what particular documentation is needed to support the application of anticancer compounds less toxic than the majority of those currently used. This paragraph reflects our professional viewpoint on everyday treatment of patients with various malignancies, hence the term “allegations” does not seem correct and carefully chosen. Using the term “novel compounds” becomes clear if the entire phrase or its crucial fragment is regarded: “…..important task of looking for novel compounds with low toxicity…”. Concerning the examples of compounds with anticancer effect resembling that of LF/8OA, some good cases with appropriate references are presented in Section 1.4 of “Introduction”. Our study was not dedicated to clinical treatment of humans hence, if the Reviewer wants some up-to-date examples of anticancer therapy, it should be understood that such are beyond the scope of the “Introduction”.

C) In line 57 the author(s) use the expression "...arrest their propagation." The propagation or the diffusion of tumor cells cannot be arrested. Can be limited or reduced or something similar. However, the author(s) should provide references supporting their statements and explanations on the meaning of the phrase. Could the author(s) please explain are there natural molecules (of which type) that can restrict tumor cells' diffusion? Example should be provided. Moreover, could the author(s) specify by "propagation" do they refer to tumor cells' diffusion or proliferation?

RESPONSE: According to the data presented by Kanwar et al., 2008 (doi: 10.1038/sj.icb.7100163), chemotherapy using various anticancer drugs in combination with LF (a natural molecule of protein type) can cause full rejection of such tumours as lung carcinoma, melanoma etc. That paper is cited in the manuscript (please, see ref.7) along with some other works presenting similar data, which is properly the example required by the Reviewer.   Concerning the term “propagation” it seems to be clear. Its meaning as stated by Webster’s New World Thesaurus is “diffusion, spread, procreation”, while Oxford Advanced Learner’s Dictionary of Current English explains the verb “to propagate” as to “increase the number of… by natural process...” and Webster’s New World Dictionary adds “to reproduce (itself), to spread”. These editions present synonyms for “proliferation” suggested by the Reviewer as “propagation, procreation, reproduction”. Hence, when we wrote that some natural molecules should arrest propagation of tumour(s), it was meant that such molecules should stop diffusion and/or proliferation of malignant cells. We will be grateful to the Reviewer for suggesting a more precise term.

D) In line 73 what do the author(s) mean by “… investigation of this phenomenon is really topical”?

RESPONSE: The authors mean exactly what is written i.e. it is high time to investigate this phenomenon.

E) Line 101-102. This sentence is confusing and should be rephrased by the author(s)

RESPONSE: The Reviewer is right. We have amended the phrase and now its meaning is quite clear: LF can be regarded as a candidate for treatment of cerebral tumours.

F) The author(s)are kindly requested to use the correct indicators for the numbers in line 160.  RESPONSE: We are thankful to the Reviewer for having noticed that error. It was corrected.

G) What is the point of parts 2.2 and 2.3? The author(s) could use one part for in vitro studies presenting all their protocols and procedures in detail and another one for in vivo animal studies. This kind of presentation the author(s) use is confusing.

RESPONSE: To our mind it is better to leave these two very short separate parts as they are. Each part contains clear statement about the source of materials (cells, animals), so any reader who might wish to reproduce an experiment will get necessary information.

H) Line 225, Concentrations, controls and samples should be mentioned in detailed by the author(s).

RESPONSE: Appropriate amendment is made.

I) Line 248, Could the author(s) explain why they used this specific concentration of LF? Is it referenced or has been studied elsewhere? Which was the injection plan the author(s) followed? Was only a single injection of 5 mg, was it repeated injections of 5mg or repeated injections of a total of 5 mg? The injection plan should be provided.

RESPONSE: This question is unclear. Nothing is written in line 248 about the concentration of LF used. Meanwhile, attentive reading of the next lines provides sufficient information about the injection plan. It is written that each mouse was receiving 5 mg of LF or LF/8OA complex. It is also explained that injections to mice started on the next day after the tumour inoculation and that such daily injections were continued for 24 days.

J) The control culture is not presented and not analyzed, which is the control culture that author(s) used? results of the normal cell evaluation are also not presented by the author(s). Moreover, the results of LF alone are not presented at all. The author(s) are kindly requested to present all of their results that they refer to.

RESPONSE: In fact, the answers to these questions are contained in the manuscript and can be obtained by attentive reading of Methods and Results. Erythrocytes, non-transformed cells, were used as control. Egress of hemoglobin from red blood cells was measured, which is an unequivocal evidence of their hemolysis, and this effect was observed more than once. The cytotoxic effect of LF is described for erythrocytes only, and it is understood why. In case of hepatoma 22a LF stimulated proliferation of malignant cells in dose-dependent manner, instead of their suppressing, which is described in the manuscript.

K) In line 322 the author(s) state that the LF-8Oa system caused "increasingly noticeable hemolysis". Are they sure of their result? Hemolysis is the rupture or destruction of red blood cells, and the author(s) state that the system they used in animal studies can cause increasingly noticeable hemolysis? The author(s) are kindly requested to present specifications on the red blood cells rupture by the LF-8OA.

RESPONSE: The authors are sure of their result and are thankful for reminding that hemolysis is the rupture or destruction of red blood cells. Figure 5 illustrates the results observed when increasing concentrations of LF/8OA complex were added to freshly isolated erythrocytes. It is seen that hemolysis indeed was “increasingly noticeable” before the concentration of LF/8OA reached ca. 20 µM (as calculated for LF) when the plot acquired the form of a saturation curve. To our opinion this graph is a good specification on the erythrocytes’ rupture.

L) Could the author(s) explain whether the animals were sacrificed at the end of the study and the tumors were collected or not? If yes could they please add photos of the tumors and dimensions for comparison reasons?

RESPONSE: This explanation is present in section 3.4 of Results:

«The changes of tumour volume in mice were continuously registered till the moment of an animal’s death». Thus, animals were not sacrificed in accordance with a preset time interval, since the aim of the study was to evaluate an increase of longevity caused by LF/8OA. Assessment of tumour growth retardation provided by LF/8OA was another important issue. Therefore, our experimental protocol previewed no withdrawal of a tumour node, since the lifetime of every mouse with hepatoma was particular and comparing the dimensions of its tumour with those extracted from other mice had not much sense.

M) Line 358, The morphology of the LF-8OA system is unclear. Could the author(s) provide more proofs and specifications on this subject?

RESPONSE: The authors have presented in the manuscript an appropriate reference of the paper by Lebedev et al., 2019 (doi: 10.1016/j.bbrc.2019.09.116) in which the “morphology” of the LF/8OA complex determined by neutron scattering is characterized in sufficient detail. It was shown that the complex is a monodisperse system with gyration radius close to that of LF, which suggests the localization of OA at the surface of LF rather than a micellar structure. Any interested reader is invited to look through that paper and acquire exhaustive information on the issue.

N) Line370-371 “It seems likely that LF-8OA is capable of forming pores in cell membrane “. How did the authors come to this conclusion?

RESPONSE: The authors came to this conclusion having observed the hemolytic effect of LF/8OA on erythrocytes.

Reviewer 3 Report

Strength:

              In this study, authors tried to clarify whether lactoferrin (LF) / oleic acid (OA)  complex has cytotoxic effect against cancer cell line. It was observed that the complex LF:OA=1:8 could show the cytotoxic effect in cultured hepatoma 22a cells. The complex LF/OA also demonstrated the hemolysis in isolated human erythrocyte in concentration dependent manner. Moreover, the complex LF/OA could decrease the size of tumor and increase the longevity in C3HA mice which were inoculated hepatoma 22a cells compared to vehicle or LF treated mice solely. These results indicated that the complex of LF/OA might be a potential tool for cancer treatment.   

              This study is interesting. I would like to ask authors about several points for this manuscript.

Major comment

  1. According to the experimental data, it seems that LF itself did not show a potent cytotoxicity against cancer cell line. In contrast, there was no data whether OA itself could induce the apoptotic effect against cancer cell line or not. Recently, it has been reported that oleic acid induced apoptosis and autophagy in the treatment of carcinomas (Jiang L et al. Sci. Rep. 7(1):11277 2017). Considering to this report, it is difficult to judge whether the cytotoxic effect induced by the complex LF/OA was brought by the complex itself without the data could demonstrate that OA itself did not have a cytotoxicity against cancer cell line, hepatoma 22a cells, at least.

Minor comments

  1. It seems that the statistical analyses have not done in all of experimental data. In particular from Fig.4 to Fig. 8, Authors should perform appropriate statistical analyses to demonstrate the cytotoxic effect of LF/OA complex.

  1. In Fig. 6 and 7, there are not error bars. Please append them.
  2. All of figures does not have any appropriate figure legends. Please append them. Number of experiments should be described particularly.
  3. In the section of materials and methods, the technical information of cultured cell experiment is not sufficient. Authors need to describe the experimental conditions and methods of cultured cell experiments more in detail.
  4. In Fig.4 the label of horizontal axis is the concentration of LF/OA complex. In contrast, the label of horizontal axis in Fig.5 is the concentration of LF. How did authors calculate and convert the concentration of LF/OA?
  5. In this study, why authors determined to examine the LF/OA complex which constituted by 1 mole of LF with 8 moles of OA? In addition, how about the cytotoxicity against cancer cell line of other complex constituted by LF with other non-saturated fatty acids?
  6. The section of introduction should be concise. The content of it is too redundant.
  7. A number of atoms in chemical formula should be described as a subscript (in page 4 at line 160).

Author Response

Major comment

  1. According to the experimental data, it seems that LF itself did not show a potent cytotoxicity against cancer cell line. In contrast, there was no data whether OA itself could induce the apoptotic effect against cancer cell line or not. Recently, it has been reported that oleic acid induced apoptosis and autophagy in the treatment of carcinomas (Jiang L et al. Sci. Rep. 7(1):11277 2017). Considering to this report, it is difficult to judge whether the cytotoxic effect induced by the complex LF/OA was brought by the complex itself without the data could demonstrate that OA itself did not have a cytotoxicity against cancer cell line, hepatoma 22a cells, at least.

RESPONSE: The Reviewer is perfectly right: our experimental data did not provide evidence of a potent cytotoxicity owned by LF. The aim of this study, as stated in “Introduction” was to combine the capacity of LF to enter malignant cells facilitating the delivery of toxic OA. We presented a reference [70] to the paper by Li et al., 2020 (doi: 10.1021/acsomega.0c01132) who compared the effect of separate fatty acids and those in complex with LF, and our results are in line with their observation. Preparing the complex LF/8OA included its dialysis and purification on a 0.45 micrometer filter, which eliminated non-bound OA from the milieu. Therefore, OA alone did not express its cytotoxic effect on tumour cells in our experiments, but only within LF/8OA. In the paper by Jiang et al., suggested by the Reviewer, OA was dissolved in 0.1% NaOH and 10% BSA, hence, the antitumour effect of its complex with the protein seems quite probable. In fact, hydrophobic fatty acids cannot be dissolved in water solutions (which was the likely reason for adding BSA in the study published by Jiang et al.), therefore, we would be grateful to the Reviewer if he/she kindly suggests a reliable method to assess the precise amount of a fatty acid in water solution in order to collect data on cytotoxic effect of OA itself.

Minor comments

  1. It seems that the statistical analyses have not done in all of experimental data. In particular from Fig.4 to Fig. 8, Authors should perform appropriate statistical analyses to demonstrate the cytotoxic effect of LF/OA complex.

RESPONSE: We are grateful for this comment, even though an attentive eye can spot the error bars in the graphs plotted in Fig.4 and Fig. 5. Similar designations have been added to other figures mentioned by the Reviewer. Statistical elaboration was improved. Comparing three survival curves in Fig. 8 showed that they differ with p=0.016 (c2=13.705). Gehan's Wilcoxon Test applied to compare the survival curves obtained for control and LF/8OA groups showed significant differences between them (p=0.00271). The same test indicated significant differences also between the curves obtained for LF and LF/8OA groups (p=0.00167). 

2. In Fig. 6 and 7, there are not error bars. Please append them.

RESPONSE: These figures illustrate gradual increase of the tumours’ dimensions. Hence, the bars will reflect error margins. Appropriate amendments are made.

3. All of figures does not have any appropriate figure legends. Please append them. Number of experiments should be described particularly.

RESPONSE: Appropriate amendments are made.

4. In the section of materials and methods, the technical information of cultured cell experiment is not sufficient. Authors need to describe the experimental conditions and methods of cultured cell experiments more in detail.

RESPONSE: Appropriate amendments are made.

5. In Fig.4 the label of horizontal axis is the concentration of LF/OA complex. In contrast, the label of horizontal axis in Fig.5 is the concentration of LF. How did authors calculate and convert the concentration of LF/OA?

RESPONSE: The ratio LF:OA was set in course of the complex preparation and after its dialysis and filtration was confirmed using a commercial kit for fatty acids measurements. All figures have been brought to the same style, so that abscissa now contains LF concentrations.

6. In this study, why authors determined to examine the LF/OA complex which constituted by 1 mole of LF with 8 moles of OA? In addition, how about the cytotoxicity against cancer cell line of other complex constituted by LF with other non-saturated fatty acids?

RESPONSE: As explained in “Introduction”, our research was rooted in the report by Fang et al., 2014 (doi: 10.1016/j.bbalip.2013.12.008) who showed the prominent capacity of the complex formed by oleic acid and lactoferrin to suppress tumour growth. 8 moles of OA is the maximum amount remaining in solution in the presence of LF. In view that other fatty acids tested in our study bound to LF at lower molar ratios, those were not of much interest to us and their anticancer effect was not explored. 

7. The section of introduction should be concise. The content of it is too redundant.

RESPONSE: On principle the Reviewer is right. We have reduced the initial size of Introduction and its content as it was presented to the Journal is ca. 2.5 times shorter. To our regret we cannot proceed like that, since omitting more fragments will make this section hardly comprehensible to a reader.

8.  A number of atoms in chemical formula should be described as a subscript (in page 4 at line 160).

RESPONSE: An appropriate amendment was introduced.

Round 2

Reviewer 2 Report

Journal: MDPI - Materials

Manuscript

Title:  " Interaction of lactoferrin with unsaturated fatty acids: in vitro and in vivo study of human lactoferrin/oleic acid complex cytotoxicity"

Author(s): Anna Elizarova, Alexey Sokolov, Valeria Kostevich, Ekaterina Kisseleva, Evgeny Zelenskiy, Oleg Panasenko, Alexander Budevich, Igor Semak, Vladimir Egorov, Giulia Pontarollo, Vincenzo De Filippis and Vadim Vasilyev

Reviewer Comments to Author(s)

Recommendation: Accepted

After a detailed evaluation of the manuscript and author’s reply after revision all the issues and questions have been addressed by the author(s) and the manuscript can be accepted for publication.

Author Response

We are thankful to the Reviewer for the high evaluation of our work

Reviewer 3 Report

Comment:

              It seems that authors responded some of my concerns appropriately. However, several points indicated were not modified seriously with excuse such as . In particular, I still have not convinced by the authors response against the cytotoxicity of OA on hepatoma 22a . I just requested authors to do the control experiment to assess the effect of LF/OA complex as precise as possible. Indeed, the cancer cell line used by Li et al. was different with the cancer cell line was used in this present study. In this regard, it should be needed to check the effect of OA itself against hepatoma 22a cells. If nobody will propose a method to assess the precise amount of a fatty acid in water solution in order to collect data on cytotoxic effect of OA itself, authors should set the experimental condition considering performing control (positive or negative) experiment such as showing the effects of LF, OA and LF/OA by deducting the effect of dissolution assistance reagents. Otherwise, it should be better to show the data and description that can prove that LF/OA complex did not contained free OA.

Author Response

In accord with a request of the Reviewer a new experiment was accomplished featuring gel filtration of the complex formed by LF and OA. The results showed quite explicitly that OA is not liberated from the complex. In another experiment DMSO was added as a solvent to OA, which allowed adding it without LF to hepatoma cells and assessing its cytotoxicity. The effect produced by OA was much less in comparison with that of LF-8OA complex. Overall the effects provided separately by  LF or OA are so weak that tumour suppression caused by LF-8OA complex cannot be explained by their summation. All the additional experiments are now described in the text with appropriate illustration and discussion. 
